# Association of Basic Psychological Need Fulfillment and School Happiness with Obesity Levels and Intensity of Physical Activity during Physical Education Classes in South Korean Adolescents

**DOI:** 10.3390/healthcare12010040

**Published:** 2023-12-23

**Authors:** Ju-Seok Yun, Gyu-Il Lee, Bo-Ram Kim

**Affiliations:** 1Department of Physical Education, Korea University, Seoul 02841, Republic of Korea; tennis2667@korea.ac.kr; 2Department of Physical Education, Kyungpook National University, Daegu 41566, Republic of Korea; mauri94@knu.ac.kr

**Keywords:** school happiness, basic psychological need fulfillment, intensity of physical activity, obesity level, moderate-to-vigorous physical activity

## Abstract

This study examined the relationship between basic psychological need fulfillment (BPNF) and school happiness in relation to the intensity of physical activity exhibited by adolescents during physical education (PE) classes and their obesity levels. We measured BPNF and school happiness using a questionnaire administered to 600 junior high school students. After exclusion, 532 questionnaires were analyzed. A 3D accelerometer (GT3X model) from Actigraph and the body mass index were used to measure physical activity intensity and obesity, respectively. The findings of this study indicate that BPNF and school happiness were significantly associated with physical activity intensity. Specifically, students who engaged in high-intensity physical activity exhibited higher levels of BPNF and school happiness. Additionally, BPNF and school happiness were not significantly related to obesity, whereas a significant relationship was observed between BPNF, school happiness, and the interaction between physical activity intensity and obesity level. This finding implies that adolescents with higher physical activity intensity and lower obesity reported higher levels of BPNF and school happiness. These findings suggest that physical activity intensity during PE classes significantly influences emotional development. Therefore, incorporating moderate-to-vigorous physical activity into PE may serve as a strategy for promoting emotional development in adolescents.

## 1. Introduction

Obesity is a significant contributor to adult diseases, such as heart disease, high blood pressure, and diabetes [1]. Obesity occurs when caloric intake is higher than expenditure [1]. In today’s modern society, industrialization and mechanization have led to sedentary lifestyles, causing an increase in obesity. Medical experts frequently emphasize the need to increase physical activity and improve dietary habits to prevent and treat obesity [2]. Physical activity has been shown to positively impact health and well-being by preventing conditions such as obesity, heart disease, high blood pressure, diabetes, depression, and various other health problems [3,4]. Physical activity must be practiced throughout the human life cycle from childhood and adolescence to adulthood and older age [1].

During adolescence, physical activity not only offers physical health benefits, such as weight maintenance and obesity prevention, but also supports mental health and academic performance, laying a strong foundation for physical activity and health in adulthood [5,6,7]. Conversely, adolescent obesity has been negatively associated with health and lifestyle outcomes in adulthood [8,9,10]. Obesity is a significant predictor of physical activity levels in PE classes because obese students are less likely to engage in physical activity during PE and exhibit lower physical fitness levels and poorer motor skills than their peers [11,12]. In addition, obese students are less likely to find satisfaction in PE classes and are more likely to experience emotional disturbances, including a lack of BPNF and lower levels of school happiness [13,14,15]. Therefore, it is important to develop teaching strategies and programs for obese students in PE curricula [15,16].

For healthy physical development, teens need a variety of forms of physical activity, including aerobic and resistance training [17]. Moderate-to-vigorous physical activity (MVPA) is a key indicator of adolescent health development, and participation in intense physical activity is more important than the total amount of physical activity [18]. The World Health Organization (WHO) recommends 60 min of MVPA per day for adolescents [19], while the National Association for Sport and Physical Education (NASPE) recommends allocating at least 50% of PE class time to MVPA [20]. However, studies have shown that more than 80% of adolescents fail to meet the recommended levels of physical activity, and their fitness levels are lower than those in the past [21].

Globally, adolescent happiness has declined in recent years. South Korean adolescents rank lowest in happiness among the 22 Organisation for Economic Co-operation and Development (OECD) countries [22]. The Adolescent Happiness Index plays a positive role in identifying adolescent health behaviors. Higher self-reported happiness is associated with a lower risk of deviant behaviors such as Internet addiction, drinking, smoking, and drug use and is inversely related to suicide rates [23]. Additionally, a happy outlook formed during adolescence continues into adulthood [24]. Among the OECD countries, South Korean adolescents rank lowest in happiness, with suicide being the leading cause of adolescent death [25]. Adolescents spend most of their time in school. School happiness, which is defined as the state of psychological satisfaction and emotional well-being experienced by adolescents during school and school-related activities, is highly correlated with overall life satisfaction [22,26]. School happiness positively impacts adolescents’ interpersonal relationships with peers and teachers as well as their cognitive performance at school. Students with higher levels of school happiness are more likely to actively engage in schoolwork and school life [27]. Therefore, school happiness has a significant influence on the Adolescent Happiness Index, and efforts should be directed toward enhancing school happiness to improve adolescents’ overall happiness levels.

All adolescents have basic, universal psychological needs. According to self-determination theory, humans have three basic psychological needs: autonomy, competence, and relatedness [28]. These needs work together organically to create an internal motivation for individuals to learn, grow, and develop [28]. Individuals whose basic psychological needs are adequately fulfilled in childhood are more likely to develop prosocial behaviors [29]. Additionally, basic psychological need fulfillment (BPNF) results in higher internal motivation which, in turn, leads to better academic achievement and greater satisfaction with school life [28,29]. Students who are satisfied with their school experience are more likely to succeed in life outside of school [30,31]. Therefore, it is necessary to pay attention to the basic psychological needs of adolescents in school because their psychological growth is significantly influenced by their schooling experience.

Numerous studies have attested to the psychological benefits of physical activity among adolescents. For example, physical activity can help prevent depression and enhance psychological well-being and happiness [32]. Studies on physical education (PE) have explored the relationship between physical activity and mental health, highlighting the psychological benefits of physical activity [33,34,35,36]. PE classes account for the majority of physical activity during school days and have a positive impact on the physical, mental, and cognitive development of adolescents [37,38]. As a result, several studies have focused on the relationship between PE and psychological factors, such as BPNF and school happiness. However, previous studies have either compared participation in PE classes or utilized unidimensional measures of physical activity through self-reported questionnaires or pedometers [39,40,41]. None of these studies have focused on the actual intensity of engagement in physical activities. The use of questionnaires to measure daily physical activity may not provide reliable results because the responses are based on the respondent’s memory [42,43]. Therefore, further research is required to develop more detailed and accurate measures of physical activity among adolescents.

To address these issues, a three-dimensional accelerometer measures human movement in three directions to calculate the minutes of physical activity by intensity (inactive and low, medium, and high intensity) over a set period. This method offers higher validity and reliability than other physical activity measurement tools and is not disruptive during classroom activities because it can be worn around the waist [44,45]. This study aimed to measure the amount of physical activity by intensity during PE classes using a three-dimensional accelerometer to obtain more accurate data on physical activity and examine its relationship with psychological factors in adolescents.

The literature does not report any studies analyzing the psychological factors associated with obesity and physical activity in the context of physical activity intensity in PE classes. We investigated the differences and direct relationships between variables such as physical activity intensity in PE classes, obesity, basic psychological needs, and school happiness. Unfortunately, no study has explored the relationship between BPNF and school happiness considering participation levels according to physical activity intensity and obesity status. Therefore, this study specifically examined the differences in adolescents’ basic psychological needs and school happiness as interaction effects between variables by categorizing the level of physical activity intensity in PE classes and the level of obesity. Consequently, the research hypotheses are as follows: First, the basic psychological needs of adolescents differ depending on the intensity of physical activity and obesity levels in PE classes. Second, adolescents’ school happiness differs depending on the intensity of physical activity and obesity levels in PE classes. Third, adolescents’ basic psychological needs and school happiness differ depending on the interaction between the intensity of physical activity and obesity levels in PE classes.

## 2. Methods

### 2.1. Participants and Sampling

Participants in this study were selected using convenience sampling from middle schools in Korea. Specifically, the sample size (n= 307) was derived using a sample size calculator (https://www.surveymonkey.com/mp/sample-size-calculator, accessed on 2 August 2018) for the population (n = 1519) enrolled in five middle schools, and a total of 600 questionnaires were distributed by securing more samples to increase validity and consistency. After that, 532 copies were used for the final analysis, excluding 68 questionnaires that were judged to be unreliable due to double entries or no entry. Measurements of physical activity intensity and surveys were conducted from the end of September to December 2018. Before commencing the study, the objectives and purpose were fully explained to the students, their homeroom teachers, and their parents, and their consent was obtained. Accordingly, measurements and surveys were conducted with students who voluntarily expressed their intention to participate. Table 1 presents the socio-demographic characteristics of the 532 participants included in the final analysis.

### 2.2. Validity and Reliability of Measurement Tools

To achieve the purpose of this study, two measurement tools were utilized: a 3D accelerometer developed by ActiGraph (Pensacola, FL, USA) and a questionnaire. First, a 3D accelerometer (GT3X model) was used to measure the physical activity intensity of students participating in PE classes at five middle schools in Korea. Three-dimensional accelerometer measurements were obtained once per class. The researcher entered the subject’s height and weight into the 3D accelerometer and configured the machine according to class time, using the height and weight information obtained from the PE teachers. The accelerometers were placed 3 cm below the students’ navels during recess before class, with instructions for students to avoid removing the accelerometers during class and to participate in the PE class as usual. At the end of the class, the data measured by the accelerometer were entered into the physical activity analysis program provided by ActiGraph, and the time and proportion of physical activity categorized by intensity during the 45 min class were analyzed.

The height and weight of the study participants were used to measure obesity. The body mass index (BMI), which is widely used to indicate an individual’s obesity status, was calculated as weight (kg) divided by height squared (m^2^) [46]. Generally, individuals are categorized as underweight (<20.0 kg/m^2^), normal weight (20.0–24.9 kg/m^2^), overweight (≥25.0 kg/m^2^), or obese (30.0+ kg/m^2^) based on their BMI levels [47]. The World Health Organization (WHO) defines obesity as a body mass index of 30 kg/m^2^ or more, and the World Health Organization Western Pacific Region (WPRO) defines obesity in Asians as a body mass index of 25 kg/m^2^ or more [2]. Due to the different criteria for diagnosing obesity using the body mass index, this study applied obesity criteria suitable for Korean adolescents and categorized them as underweight (<20.0 kg/m^2^), normal weight (20.0–24.9 kg/m^2^), and obese (BMI ≥ 25 kg/m^2^).

A questionnaire was administered to collect information on demographic variables, basic psychological needs, and school happiness. To measure basic psychological needs, we used a scale developed by Choi, which was also employed by Kang and Cho [48,49]. Additionally, this scale was validated by Kang and Cho [49]. Specifically, the scale consists of thirteen questions with three subfactors: autonomy, competence, and relatedness. An example item is “I can freely express my opinions and thoughts in PE class”. The questions were rated on a five-point Likert scale (1 = not at all; 5 = very much).

The School Happiness Scale, developed by Kang, was used in a previous study by Kang and Cho [50,51]. Additionally, this scale was validated by Kang and Cho [51]. This scale consists of seventeen questions related to four subfactors: self-esteem, optimism, relationships with friends, and relationships with teachers. Examples of items include “I do well in everything I do at school” and “I think of others first at school”. Responses were recorded on a five-point Likert scale (1 = not at all; 5 = very much). The survey questions are presented in Table 2.

Based on this, an expert meeting comprising two professors and two doctorates in physical education reviewed whether the survey contents were suitable for middle school students. Additionally, a final check was conducted on 100 middle school students for understanding, appropriateness, and expected time spent on the survey questions. Subsequently, the survey questions were revised and supplemented to obtain the final questionnaire.

Next, a confirmatory factor analysis was conducted to verify construct validity between the potential variables and measurement variables of basic psychological needs and school happiness. The suitability evaluation criteria were confirmed based on absolute fit indices, including the CMIN/DF, RMSEA, SRMR, and GFI, and incremental fit indices (CFI, NFI, and TLI) [52]. The analysis showed that the basic psychological needs met the criteria with χ^2^ = 107.062, df = 37, RMSEA = 0.059, SRMR = 0.029, GFI = 0.971, CFI = 0.989, NFI = 0.983, and TLI = 0.976. School happiness also met the standards with χ^2^ = 254.201, df = 85, RMSEA = 0.060, SRMR = 0.036, GFI = 0.950, CFI = 0.964, NFI = 0.947, and TLI = 0.942 (Table 3). Cronbach’s α was calculated to verify the reliability of the measurement tools. Cronbach’s α values were 0.945 for basic psychological needs and 0.921 for school happiness, ensuring high reliability.

### 2.3. Data Processing Method

SPSS 27.0 and AMOS 25.0 were used for data processing. A frequency analysis was conducted to identify the general characteristics of the participants. A confirmatory factor analysis was conducted to verify the dimensionality and validity of the factor structure of the variables, and Cronbach’s α coefficient was calculated to verify reliability. A K-means cluster analysis was performed to classify the level of physical activity intensity. Finally, *t*-tests, a one-way ANOVA, and a two-way ANOVA were used to test the hypotheses.

## 3. Results

### 3.1. Cluster Analysis

To classify the level of physical activity intensity, a K-means cluster analysis was conducted based on the MVPA index (Table 4). As a result of the analysis, participants were classified into two groups, upper (M = 15.29) and lower (M = 6.20), with a significant difference (*F* = 1171.322, *p* < 0.001) observed between the groups (Table 4).

### 3.2. Fulfillment of Basic Psychological Needs According to Physical Activity Intensity and Obesity

A t-test and one-way ANOVA were conducted to test for differences in BPNF according to physical activity intensity and obesity. The analysis revealed a statistically significant difference (*t* = 5.284, *p* < 0.001) in BPNF according to physical activity intensity. Specifically, the high physical activity intensity group (M = 3.70) had a higher BPNF than the low physical activity intensity group (M = 3.34) (Table 5). There was no statistically significant difference in BPNF according to the obesity level (Table 6).

### 3.3. School Happiness According to Physical Activity Intensity and Obesity

A *t*-test and one-way ANOVA were conducted to investigate differences in school happiness according to physical activity intensity and obesity status. The analysis revealed a statistically significant difference (*t* = 3.870, *p* < 0.001) in school happiness based on physical activity intensity. Specifically, the high-intensity physical activity group (M = 3.51) demonstrated higher school happiness than did the low-intensity physical activity group (M = 3.28) (Table 7). There was no statistically significant difference in school happiness according to the obesity level (Table 8).

### 3.4. Basic Psychological Needs According to the Interaction between Physical Activity Intensity and Obesity

Two-way ANOVAs were conducted to examine the differences in BPNF based on the interaction between physical activity intensity and obesity level (Table 9, Figure 1). Differences in BPNF among the six groups based on six (2 × 3) combinations of physical activity intensity and obesity level were investigated. An analysis of the interaction effect between physical activity intensity and obesity revealed a statistically significant difference (*F* = 5.039, *p* < 0.01). Specifically, underweight students with a high level of physical activity intensity (M = 3.93) exhibited the highest satisfaction with their basic psychological needs, followed by students with a high level of physical activity intensity and normal weight (M = 3.64). In contrast, students with low physical activity intensity and obesity (M = 3.25) reported the lowest satisfaction with basic psychological needs, followed by students with low physical activity intensity and underweight status (M = 3.27). Additionally, the main effect analysis indicated a significant difference in BPNF according to physical activity intensity (*F* = 28.252, *p* < 0.001) but no significant difference in satisfaction with basic psychological needs according to obesity level (*F* = 2.667, *p* = 0.07).

### 3.5. School Happiness According to the Interaction between Physical Activity Intensity and Obesity

To explore differences in school happiness based on the interaction between physical activity intensity and obesity level, a two-way ANOVA was conducted (Table 10 and Figure 2). Differences in school happiness among the six groups (2 × 3) categorized by physical activity intensity and obesity level were also investigated. Upon analyzing the interaction effect of physical activity intensity and obesity, the findings revealed a statistically significant difference (F = 5.678, *p* < 0.01). Specifically, underweight students with a high physical activity intensity (M = 3.73) exhibited the highest level of school happiness, followed by normal-weight students with a high physical activity intensity (M = 3.42). Contrastingly, students with low physical activity intensity and obesity (M = 3.20) had the lowest level of school happiness, followed by underweight students with low physical activity intensity (M = 3.24).

Additionally, the main effect analysis indicated a significant difference in school happiness according to physical activity intensity (F = 16.737, *p* < 0.001) but no significant difference in school happiness according to obesity level (F = 2.855, *p* = 0.06).

## 4. Discussion

This study investigated the relationship between basic psychological needs and school happiness in relation to the intensity of physical activity and obesity levels among students in PE classes. We present the following findings and implications based on the results of this study.

### 4.1. Interpretation of Findings

First, there was a significant difference in BPNF in PE students according to physical activity intensity, whereas no differences were observed in BPNF according to obesity levels. The relatively high-intensity physical activity group demonstrated higher BPNF than the low-intensity physical activity group. This finding aligns with those of previous studies, indicating that BPNF tends to increase with the duration of physical exercise. According to Eum and Ko, students who participated more frequently in school sports clubs and thus exercised more often tended to feel more competent and autonomous. Those who exercised 3–4 times a week were also more likely to establish relationships [53]. In other words, frequent and extended physical activity is associated with higher levels of baseline psychological motivation [28]. MVPA, the intensity of the physical activity measure used in this study, was also found to be significantly related to the total time spent exercising. For example, MVPA is higher on days with access to gymnasiums, indicating higher overall physical activity levels [54]. This indicates a genuine increase in physical activity without a corresponding decrease in other areas (such as sedentary lifestyle and low-intensity physical activity) [55]. In this study, individuals with relatively high MVPA engaged in more moderate and vigorous exercise than those with low MVPA. However, they devoted more time to total physical activity than those with low MVPA. Therefore, the results of the present study support those of previous studies [53,56], suggesting that increased time spent engaging in physical activity is associated with higher BPNF.

However, there was no difference in BPNF levels across obesity levels. According to the self-determination theory, humans have three basic psychological needs (autonomy, relatedness, and competence), and the degree to which they are satisfied is linked to higher motivation levels [28]. Autonomy in classroom situations refers to the degree to which students perceive that they have choices and are adequately represented; for example, how autonomous they feel in their own learning [57]. Competence refers to one’s desire to grow, learn, and develop, reflecting an individual’s confidence in their ability to perform effectively; for example, when they successfully complete a task to meet their own standards. Finally, relatedness concerns establishing and developing lasting relationships with others and involves a sense of belonging and support through interactions [58]. School-aged students’ basic psychological needs are related to their perceived satisfaction with the school setting. Therefore, it is unlikely that a person’s physical characteristics, such as obesity, make a difference. The results of this study are consistent with the findings of Lee and Park, who found no significant relationship between basic psychological needs satisfaction and BMI [59]. Similarly, Lim and Kim found no differences in intrinsic motivation in PE classes based on obesity and physical fitness [60].

Second, a significant difference was observed in school well-being based on the intensity of physical activity among students in PE classes, whereas no differences were noted in school well-being based on obesity level. Specifically, those who were more physically active reported higher levels of school happiness than those who were less physically active. These results can also be inferred from the fact that MVPA, which represents physical activity intensity, is derived from the total time spent engaging in physical activity, as mentioned earlier. According to An et al., higher levels of physical activity tend to be associated with higher levels of life satisfaction and happiness across all age groups, including youth, adults, and older adults [32]. Additionally, Pengpid and Peltzer reported that college students who participated in MVPA reported higher life satisfaction and happiness than those who did not [61]. Lee et al. found that participation in school PE classes was positively associated with perceived subjective well-being, suggesting that more frequent participation in PE classes was associated with higher levels of perceived subjective well-being [39]. Yoo and Kim found that students who participated in more PE classes reported higher levels of happiness and lower levels of perceived stress and suicide attempts [62]. Thus, our findings are consistent with those of previous studies indicating that relatively high levels of physical activity intensity and increased physical activity participation positively impact subjective well-being.

However, there was no significant difference in school happiness according to obesity level. This suggests that there is no significant difference in subjective school happiness based on obesity level. However, although the difference was not statistically significant, obese students demonstrated a lower mean value for school happiness than normal-weight and underweight students. These findings are consistent with those of several studies that have found differences in happiness levels based on obesity levels. Choi employed three years of large-scale data from the Korean Youth Health Behavior Online Survey and found an indirect effect of BMI on happiness [63]. According to Cornelisse-Vermaat and Van, BMI has a slightly negative impact on well-being [64]. However, the effect of obesity on individual psychological factors, such as happiness, remains unclear [65,66]. Atlantis and Baker found that the association between obesity and depression varied across countries [67]. The stronger the negative perception of obesity, social prejudice, and discrimination based on physical appearance, the greater the impact on the psychosocial aspects of the individual [68,69]. Conversely, Swallen et al. found that being overweight and obese negatively impacted depression, self-esteem, and school social functioning, and that adolescent obesity was associated with lower quality of life [70]. Previous studies demonstrated an inverted U-shaped relationship between obesity and happiness [71]. Overall, conflicting studies on obesity and its psychosocial causation have made it difficult to draw firm conclusions [65,66]. Thus, our finding of no significant difference in school happiness in relation to obesity level is consistent with the growing body of research on the psychosocial relationship between obesity and school well-being.

Third, the interaction between physical activity intensity and obesity level demonstrated significant differences in BPNF and school happiness for each variable. For basic psychological needs, students with high physical activity levels and low body weight reported the highest basic psychological need fulfillment, followed by students with high physical activity levels and normal body weight. In contrast, students who were physically inactive and obese had the lowest basic psychological need fulfillment, followed by those who were physically inactive and underweight. In essence, students who were physically more active and less obese tended to have higher levels of basic psychological need fulfillment. In the case of school happiness, students with high physical activity levels and low body weight had the highest levels of school happiness, followed by students with high physical activity and normal body weight. In contrast, students who were physically inactive and obese had the lowest levels of school happiness, followed by those who were physically inactive and underweight.

Both BPNF and school happiness differed according to physical activity intensity and obesity, with the highest scores observed in the more physically active and non-obese groups. Contrastingly, all groups in the bottom quartiles of BPNF and school happiness demonstrated low physical activity intensity. These results suggest that physical activity intensity is more influential than obesity among adolescents and that physical activity intensity in PE classes is a major contributor to BPNF and school happiness. As mentioned earlier, high physical activity intensity can positively affect both BPNF [28] and school happiness [61]. In particular, increased physical activity in PE class contexts is directly related to students’ physical and psychological health as well as their overall life satisfaction and happiness [39,72]. Therefore, strategies should be explored within PE classes to achieve and actively promote the recommended daily physical activity levels for adolescents (60 min of MVPA) for healthy physical development and emotional development.

However, the current findings have several implications for students with low physical activity levels and those ranking last in obesity. Obese youth in Korea face risks related to various physical and mental health issues, including cardiovascular disease, anxiety, and depression [73]. Therefore, robust physical activity is necessary for obese adolescents as participation in exercise promotes vitality and self-esteem while mitigating negative states such as depression and tension [74]. However, the more obese individuals are, the more likely they are to experience difficulties engaging in physical activity, creating a vicious cycle [75,76,77]. Therefore, it is necessary to take an active interest in supporting these students to lead healthier and happier lives in schools, which are public institutions. PE classes offer an ideal platform for promoting physical activity, and PE teachers must apply diverse strategies and methods to motivate students to become physically active.

### 4.2. Practical Implications

Based on the findings of this study, several practical implications can be determined regarding physical activity intensity in PE classes for obese students.

First, MVPA should be promoted in PE classes to not only maintain students’ physical fitness but also for their emotional development. MVPA during adolescence contributes to healthy physical, psychological, and interpersonal development as well as the prevention of adult-onset diseases and psychological issues such as anxiety and depression. The WHO recommends at least 60 min of MVPA to maintain and improve adolescent health [19]. The NASPE also recommends allocating at least 50% of the PE class duration to MVPA [20]. This study also confirmed that MVPA in PE classes has a positive impact on emotional development, including satisfaction with basic psychological needs and happiness. Particularly, PE classes account for the majority of physical activities during school days [78] and have a positive impact on the physical, mental, and cognitive development of adolescents [37,38]. Additionally, MVPA was higher on days with PE classes, positively affecting heart rate [79,80]. Therefore, PE plays an important role in adolescents’ physical and mental health development, helping them achieve recommended MVPA levels according to their age. Unfortunately, more than 80% of adolescents do not meet the recommended levels of physical activity, and studies continue to indicate declining fitness levels [21]. Furthermore, PE classes do not typically provide the required amount of MVPA. PE classes that ignore adolescent health development present a challenge for justifying school physical education. Health education must not be overlooked if PE is to become an integral part of school curriculum [81]. While this study has thus far focused on the quantitative expansion of PE time in the school curriculum, attention must shift toward the qualitative expansion of activities covered in PE classes. Meta-analyses on the current state of MVPA in PE generally suggest the implementation of effective physical activity promotion strategies to increase MVPA in PE [82,83]. The development of PE curricula that incorporate these physical activity promotion strategies and their implementation in schools are important aspects of adolescent health.

Second, to improve the mental health of obese students, it is important to combat their stigmatization and actively provide MVPA in PE classes. Overweight or obese students typically exhibit low levels of physical performance and fitness, experience serious health problems, and suffer psychosocial and emotional harm from peer teasing and exclusion [15]. Additionally, these students are often perceived as unmotivated or lacking sports skills and abilities [84], leading to their exclusion from sports and PE activities [75,76]. These negative stereotypes can be internalized by overweight and obese students, causing them to avoid PE and physical activity, which can lead to additional weight gain [77]. In this study, obese students who participated in high- and medium-intensity physical activities during PE demonstrated higher levels of basic psychological need fulfillment and school happiness than those who participated in low-intensity physical activities. These findings suggest that paying attention to overweight and obese students and participating in MVPA programs can improve their emotional development. However, PE teachers are often biased toward overweight or obese students [85]. Negative stereotyping by teachers can lead to low levels of enjoyment, avoidance of PE classes, and the marginalization of these students. Therefore, it is important for PE teachers to avoid assuming that obese students do not benefit from moderate-to-high-intensity exercise. Providing MVPA in PE classes that are appropriate for their needs can not only improve the psychological and emotional well-being of overweight and obese students but also provide meaningful education to their peers.

## 5. Conclusions

This study examined the relationships among BPNF, school happiness, physical activity intensity, and obesity levels among adolescent students in PE classes. We found significant differences in BPNF and school happiness in relation to the intensity of students’ physical activity in PE classes. Specifically, those who engaged in relatively high-intensity physical activity demonstrated higher levels of BPNF and school happiness than those who engaged in low-intensity physical activity. However, no significant differences were observed in BPNF or school happiness in relation to obesity level. Therefore, obesity was not associated with psychological factors in adolescents. Notably, the findings indicated significant differences in BPNF and school happiness in relation to the interaction between physical activity intensity and obesity. Participants who were more physically active and less obese exhibited higher levels of BPNF and school happiness. In other words, the intensity of the students’ physical activity during PE classes had a significant effect on their emotional development. Therefore, strategies to promote physical activity should be explored and actively utilized, emphasizing that MVPA in PE classes significantly benefits the physical and emotional development of adolescents.

Finally, this study has the following limitations. First, we were unable to classify subjects into high- and low-physical-activity groups using the absolute standard of MVPA. This study used a cluster analysis to classify subjects into relatively high and low groups based on the midpoint of the mean MVPA of the subjects. Future studies should use the recommended standard of MVPA recommended by the WHO to classify groups. Second, we used the BMI to classify obesity into three groups: underweight, normal weight, and obese. We would have liked to use body fat percentage to measure obesity in this study, but due to the large number of subjects and a lack of equipment, we used the BMI, which is a simple measurement. The use of the BMI has been challenged in the past because it relies on height and weight to determine the level of obesity, and it is difficult to set a standard for the BMI that can cover all regions of the world. In future studies on adolescent obesity and physical activity, it is recommended to use body fat percentage to categorize groups.

## Figures and Tables

**Figure 1 healthcare-12-00040-f001:**
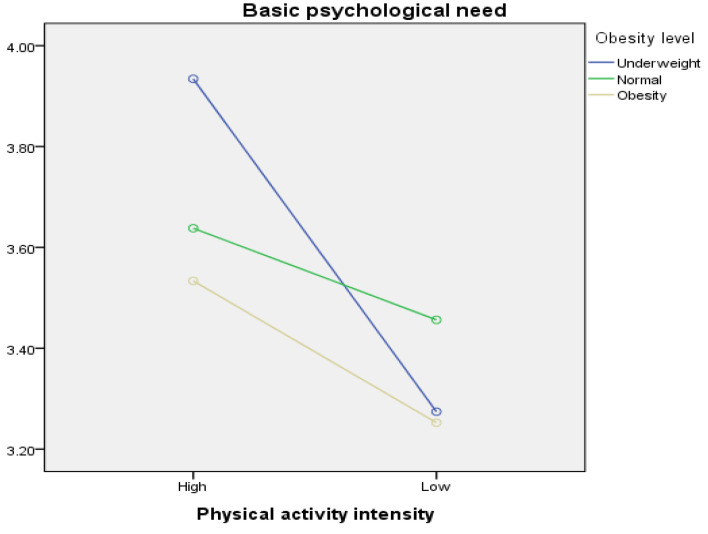
Basic psychological needs according to physical activity intensity and obesity.

**Figure 2 healthcare-12-00040-f002:**
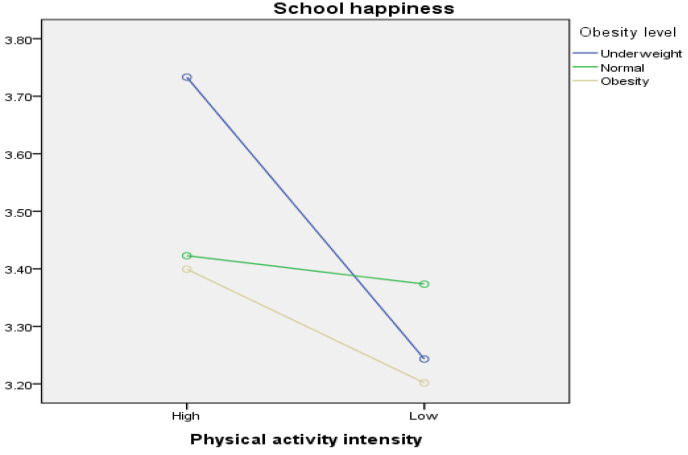
School happiness according to physical activity intensity and obesity.

**Table 1 healthcare-12-00040-t001:** Demographic characteristics.

Variable	Category	*n*	%
Gender	Male	270	50.8
	Female	262	49.2
Physical activity intensity	High intensity	204	38.3
	Low intensity	328	61.7
Obesity level (BMI kg/m^2^)	Underweight < 20.0	193	36.3
	Normal 20.0–24.9	228	42.9
	Obesity ≥ 25.0	111	20.9
Variable	Mean ± SD	n	%
Age (year)	13.57 ± 0.68	532	100.0
Height (cm)	162.05 ± 7.17	532	100.0
Weight (kg)	58.54 ± 20.84	532	100.0

**Table 2 healthcare-12-00040-t002:** Survey composition.

Category (Number of Questions)		Example of Questions
Demographic characteristics		Gender, Age, Height, and Weight
Basic psychological needs (13)	Autonomy (4)	- I can freely express my opinions and thoughts in PE class. - There are ample opportunities to exercise freely in my own way. - PE class suits my interests. - I feel a lot of freedom in PE class.
	Competence (4)	- I am not very good at sports. (R) - I tend to achieve my goals in PE classes quickly. - I feel very competent when it comes to PE. - I think I am good at PE.
	Relationship (5) (relatedness)	- In PE class, I seem to connect well with my friends. - My friends who are in PE class together are very precious. - In PE class, I feel very close to my friends. - I get along well with my friends in PE class. - I feel safe when I am with my friends during PE class.
School happiness (17)	Self-esteem (6)	- I do my job well. - I do well in everything I do at school. - I am smart. - I get recognition from friends and teachers at school. - I am confident in everything I do at school. - I am popular with my friends at school.
	Optimism (5)	- I share or lend books to friends who do not bring textbooks or materials. - I feel at ease when I come to school. - I have a friend at school with whom I can talk comfortably. - There are many times when I am happy at school. - I am not ignored at school.
	Relationships with friends (3)	- I don’t fight with other people at school. - I think of others first at school. - When I get into a quarrel at school, I am the first to apologize.
	Relationship with teacher (3)	- I easily talk about my feelings with my teacher. - I enjoy playing or talking with teachers. - I always feel free to say what I want to say to the teacher.

R: Reverse question.

**Table 3 healthcare-12-00040-t003:** Fit of the measurement tool.

Variables	χ^2^/df	RMSEA	SRMR	GFI	CFI	NFI	TLI	Cronbach’s α
Standard	≤3.0	≤0.08	≤0.08	≥0.90	≥0.90	≥0.90	≥0.90	>0.70
Basic psychological needs	107.062/37 (2.89)	0.059	0.029	0.971	0.989	0.983	0.976	0.945
School happiness	254.201/85 (2.99)	0.060	0.036	0.950	0.964	0.947	0.942	0.921

**Table 4 healthcare-12-00040-t004:** Cluster analysis of MVPA.

Variables	Intensity Level		*F*
	High	Low	
MVPA	15.29	6.20	1172.322 ***
n	204	328	

**** p* < 0.001.

**Table 5 healthcare-12-00040-t005:** Basic psychological needs according to physical activity intensity.

Variable		n	M	SD	*t*	ES
**Physical activity intensity**	High	204	3.70	0.77	5.284 ***	0.474
	Low	328	3.34	0.75		

**** p* < *0*.001; ES: Cohen’s effect size, d.

**Table 6 healthcare-12-00040-t006:** Basic psychological needs according to obesity level.

Variable		n	M	SD	*F*	*η* ^2^	Post Hoc
**Obesity level**	Underweight	193	3.48	0.86	1.593	0.006	NS
	Normal	228	3.53	0.74			
	Obesity	111	3.37	0.69			

**Table 7 healthcare-12-00040-t007:** School happiness according to physical activity intensity.

Variable		n	M	SD	*t*	*ES*
**Physical activity intensity**	High	204	3.51	0.66	3.870 ***	0.356
	Low	328	3.28	0.63		

**** p* < 0.001; ES: Cohen’s effect size, d.

**Table 8 healthcare-12-00040-t008:** School happiness according to obesity level.

Variable		n	M	SD	*F*	*η* ^2^	Post Hoc
**Obesity level**	Underweight	193	3.40	0.61	1.272	0.005	0.005
	Normal	228	3.40	0.67			
	Obese	111	3.28	0.69			

**Table 9 healthcare-12-00040-t009:** Differences in BPNF according to the interaction between physical activity intensity and obesity level.

Physical Activity Intensity	Obesity Level		n	M	SD	
High	Underweight (①)		62	3.93	0.81	
	Normal (②)		95	3.64	0.74	
	Obese (③)		47	3.53	0.70	
	Total		204	3.70	0.77	
Low	Underweight (④)		131	3.27	0.80	
	Normal (⑤)		133	3.45	0.73	
	Obese (⑥)		64	3.25	0.66	
	Total		328	3.34	0.75	
Variable	SS	df	MS	*F*	** *η* ^2^ **	**Post hoc**
Physical activity intensity	16.020	1	16.020	28.252 ***	0.051	① > ④, ⑤, ⑥ ② > ④, ⑥
Obesity level	3.024	2	1.512	2.667	0.010	
Physical activity intensity × obesity level	5.715	2	2.857	5.039 **	0.019	
Error	298.259	526	0.567			

** *p* < 0.05, *** *p* < 0.001.

**Table 10 healthcare-12-00040-t010:** Differences in school happiness according to the interaction between physical activity intensity and obesity level.

Physical Activity Intensity	Obesity Level		n	M	SD	
High	Underweight (①)		62	3.73	0.62	
	Normal (②)		95	3.42	0.69	
	Obese (③)		47	3.39	0.58	
	Total		204	3.51	0.66	
Low	Underweight (④)		131	3.24	0.54	
	Normal (⑤)		133	3.37	0.65	
	Obese (⑥)		64	3.20	0.75	
	Total		328	3.28	0.63	
Variable	SS	df	MS	*F*	** *η* ** ** ^2^ **	**Post Hoc**
Physical activity intensity	6.900	1	6.900	16.737 ***	0.031	① > ②, ④, ⑤, ⑥
Obesity level	2.354	2	1.177	2.855	0.011	
Physical activity intensity × Obesity level	4.682	2	2.341	5.678 **	0.021	
Error	216.857	526	0.412			

** *p* < 0.05, *** *p* < 0.001.

## Data Availability

The data presented in this study are available upon request from the corresponding authors.

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
