# Peer review of "Association of Basic Psychological Need Fulfillment and School Happiness with Obesity Levels and Intensity of Physical Activity during Physical Education Classes in South Korean Adolescents"

_healthcare, 2023, doi:10.3390/healthcare12010040_

Round 1

Reviewer 1 Report

Comments and Suggestions for Authors

The article examines the differences in the basic psychological needs of adolescents and their school happiness as effects of interaction between variables in the categories of physical activity intensity in physical education classes and obesity. The authors have carried out a significant amount of work on the analysis of scientific and methodological literature, involving a significant number of participants in the pedagogical experiment. The results obtained in the course of the study can form the basis for the development of means of influence on attracting adolescents to systematic physical exercises.

The study is presented in a good structural form. The course of the pedagogical experiment is logically traced.

In terms of relevance, the author analyses the research problem from different angles, while the sources considered in the analysis were mainly not for the last five years. Modern sources (for the last five years) account for only about 17% of the total number of sources.

The plan of the pedagogical experiment corresponds to the hypotheses that the authors wanted to test. Instead, the authors should personalise the hypothesis by specifying the population. For example: First, there are differences in basic psychological needs children ….

The research methods allow us to repeat the course of the study. The authors should explain why this particular classification of Body Mass Index and physical activity level was used. The classification of Body Mass Index should be based on the classification proposed by the World Health Organization. It also raises the question of why the intensity of physical activity was studied only in physical education classes where its content is regulated by the teacher, and not the content of physical activity during the day. For example, the level of physical activity on a day with a physical education lesson, without a physical education lesson and on a day off.

The data obtained correspond to the research hypotheses planned for consideration. The presentation of the material is conducive to understanding the course of the study. The description of the data obtained does not always require their presentation in tables (Tables 4-8).

The conclusions generally reflect the data obtained and allow us to assess the degree of confirmation of the research hypothesis.

Author Response

Thank you so much for taking time out of your busy schedule to review.

Please check the attached file below for a response regarding corrections.

The revised parts of the manuscript are checked in red.

thank you

Reviewer 2 Report

Comments and Suggestions for Authors This study examined the relationship between basic psychological needs satisfaction (BPNF) and school happiness in relation to the intensity of physical activity exhibited by adolescents during physical education (PE) classes and their obesity levels. I suggest adding this quote in the introduction: https://doi.org/10.3390/ijerph17155300   The methodology is adequate, and the sample is large. As an improvement review, I need to know if the happiness and autonomy scale are validated, and who are the authors of this validation. If it is not validated, then it does not seem that way, since it states: "A questionnaire was administered to collect information on demographic variables, basic psychological needs, and school happiness. A panel of experts, comprising two physical education professors and two physicians, determined the suitability and validity of the survey questions. In addition, middle school students reviewed the survey questions for comprehensibility, appropriateness, and the expected time required to an-165 swer the survey questions, ensuring content validity. Subsequently, the survey questions 166 were revised and supplemented to obtain the final questionnaire." If the scale is not validated, I recommend redirecting the article, validating the scale and pointing out the results in the same article. Lastly, I don't understand an anova without a post hoc test.

Thanks 

Best regards 

Author Response

(The authors gave the same response as above.)

Reviewer 3 Report

Comments and Suggestions for Authors

“Association of basic psychological need fulfillment and school  happiness with obesity and intensity of physical activity during physical education classes in South Korean adolescents”( healthcare-2739864)

This manuscript aimed to explore the association of basic psychological need fulfillment and school happiness with obesity and intensity of physical activity during physical education classes in South Korean adolescents. Physical activity was measured by Actigraph objectively, which is desirable. The results revealed that students who engaged in high-intensity physical activity exhibited higher levels of basic psychological need fulfillment and school happiness. In addition, BPNF and school happiness were not significantly related to obesity, whereas a significant relationship was observed between BPNF, school happiness, and the interaction between physical activity intensity and obesity level. Overall, this topic seems interesting. However, some concerns appeared after reading the whole manuscript.

1. The literature review part is far from satisfactory, both review about the international articles and the review in Korea is not comprehensive. The reference list needs to be updated. Some important papers need to be reviewed and discussed, such as,

Kang, H. K., & Cho, J. H. (2018). Examination of Influences of Elementary Schoolers' Basic Psychological Needs in Physical Education Classes on School Happiness and Physical Lifestyles. The Journal of the Korea Contents Association, 18(10), 584-595.

Koucheki, A. M., Shariatnia, K., Asadi, A., & Mirani, A. (2022). The mediating role of psychological basic needs in the relationship between personality traits and students’ happiness. Journal of psychologicalscience, 21(118), 2091-2106.

Shin-bok, Y., & Sook, S. W. (2013). Relationships of perceived maternal emotional expressiveness with basic psychological needs and school happiness for korean elementary schoolers. The Korean Journal of School Psychology, 10(1), 179-200.

Yu-Ting, Y., Miao, Y., Yong-Wei, Y., Qiong, Y., & Ting, L. (2022). Relationships between children-related factors, basic psychological need satisfaction, and multiple happiness among urban empty-nesters in China: a structural equation modeling. BMC geriatrics, 22(1), 1-16.

Jeon, W., Lee, B., & Joung, K. (2023). A Study on the Relationships between Playfulness, Physical Selfefficacy and School Happiness among Middle School Students participating in" 0th-Period Physical Education Class" in South Korea. Frontiers in Public Health, 11, 1232508.

Lin, S., Li, L., Zheng, D., & Jiang, L. (2022). Physical exercise and undergraduate students’ subjective well-being: Mediating roles of basic psychological need satisfaction and sleep quality. Behavioral Sciences, 12(9), 316.

Vansteenkiste, M., Ryan, R. M., & Soenens, B. (2020). Basic psychological need theory: Advancements, critical themes, and future directions. Motivation and emotion, 44, 1-31.

Overall, the current manuscript seems too Korean-oriented and the authors should put their study into the more international context.

2. After reading the whole manuscript, it is still confuse me why the authors did this investigation and what are the research gaps in this field. Also the relationship between variables should be better summarized and the novelties of the current investigation need to be stated more clearly.

3. How did you determine the sample size? Did you calculate the sample size needed before formal study?

4. For the hypotheses, the current version is too vague and the directions should be pointed out in the hypotheses. Also, the rationales for the development of each hypothesis need to be provided.

5.“at the end of March and June” which year?

6. As for the School Happiness Scale, the original School Happiness Scale contains 24 items, which comprises six domains (peer relationships, relationships with teachers, self-efficacy, environmental satisfaction, pleasure in learning activities, and psychological stability). Why did you only included 17 items?

Lee, S. M., Yoo, J. I., & Youn, H. S. (2021). Changes in alienation in physical education classes, school happiness, and expectations of a future healthy life after the COVID-19 pandemic in Korean adolescents. International journal of environmental research and public health, 18(20), 10981.

7. I cannot find the reference [48] in the search

8. What method did you use for the correction of multiple comparisons?

9. What is the cutoff point of MVPA index did you get to classify high and low physical activity intensity?How did you get the MVPA index is not specified.

10. Please provide the effect size where available.

11. “***p<.001” is no need for table 6 and 8.

12. Why did you not provide the specific data for χ2/df of basic psychological needs and school happiness in table 3.

13. About the limitations, some explaination might be more desirable.

14. I strongly recommend that the paper be thoroughly proofread and edited for logic, languages and grammars, to enhance readership.

Comments on the Quality of English Language

English very difficult to understand/incomprehensible

Author Response

(The authors gave the same response as above.)

Reviewer 4 Report

Comments and Suggestions for Authors

Dear Athuros,

Thank you for your manuscript; please see the attached file for suggestions.

Author Response

Thank you very much for taking the time to review this manuscript.

Please check the detailed answer and corrections in the attached file below.

In the manuscript, the revised parts are checked in red. thank you

Round 2

Reviewer 2 Report

Comments and Suggestions for Authors

I agree with the changes made to the work, in my opinion it is sufficient for publication.

Thank you

Greetings

Author Response

Once again, I would like to express my sincere gratitude.

Thank you very much for your review to improve the quality of the paper.

Reviewer 3 Report

Comments and Suggestions for Authors

Thanks for the revisions and one minor concern remains.

Please add the sample size determination part in the formal context.

Comments on the Quality of English Language

Minor editing of English language required

Author Response

Yes, based on the reviewer's opinion, we have added the following.

Specifically, the sample size (N=307) was derived using a sample size calculator(https://www.surveymonkey.com/mp/sample-size-calculator) for the population(N=1,519) enrolled in five middle schools, and a total of 600 questionnaires were distributed by securing more samples to increase validity and consistency. After that, 532 copies were used for the final analysis, excluding 68 questionnaires that were judged to be unreliable due to double or no entry.

Once again, I would like to sincerely thank you for taking the time to improve the quality of this research despite your busy schedule.

Reviewer 4 Report

Comments and Suggestions for Authors

Dear authors,

thank you very much for your effort. Actually, the manuscript is deeply changed after these improvements. 

The introduction was deeply modified and improved. Now the level has grown up. Congratualtion!!

I ca affirm the same for the rest of the manuscript. However, I have the last suggestion for you. From line 463 to 466 you declare: ". The use of BMI has been challenged in the past because it relies on height and weight to determine the level of obesity, and it is difficult to set a standard for BMI that can cover all regions of the world. In future studies on adolescent obesity and physical activity, it is recommended to use body fat percentage to categorize groups." In these sentences, you affirm that BMI is a limit and you suggest the use of body fat percentage for future studies. However, this affirmation lead me to a question: Why did you not use the fat percentage on this study? I suggest you to explain this (the sample size, inadequate logistics, lack of certified personelle etc.). BMI is still good in large samples and in general population. It can be not signficant in well trained athletes where muscle mass is extremly developed. 

Author Response

Yes, based on the reviewer's opinion, we have added the following to the limitations of the study. thank you

"We would have liked to use body fat percentage to measure obesity in this study, but due to the large number of subjects and lack of equipment, we used BMI, which is a simple measurement. "

Once again, I would like to sincerely thank you for taking the time to improve the quality of this research despite your busy schedule.
